

# Uncertainty and sensitivity analysis of the basic reproduction number of diphtheria: a case study of a Rohingya refugee camp in Bangladesh, November–December 2017

Ryota Matsuyama[1,2], Andrei R. Akhmetzhanov[1,2], Akira Endo[1,3], Hyojung Lee[1,2], Takayuki Yamaguchi[1,2], Shinya Tsuzuki[1,2] and Hiroshi Nishiura[1,2]

[1] Graduate School of Medicine, Hokkaido University, Sapporo, Japan
[2] Japan Science and Technology Agency, Core Research for Evolutional Science and Technology, Saitama, Japan
[3] London School of Hygiene & Tropical Medicine, University of London, London, UK

## ABSTRACT

**Background:** A Rohingya refugee camp in Cox's Bazar, Bangladesh experienced a large-scale diphtheria epidemic in 2017. The background information of previously immune fraction among refugees cannot be explicitly estimated, and thus we conducted an uncertainty analysis of the basic reproduction number, $R_0$.

**Methods:** A renewal process model was devised to estimate the $R_0$ and ascertainment rate of cases, and loss of susceptible individuals was modeled as one minus the sum of initially immune fraction and the fraction naturally infected during the epidemic. To account for the uncertainty of initially immune fraction, we employed a Latin Hypercube sampling (LHS) method.

**Results:** $R_0$ ranged from 4.7 to 14.8 with the median estimate at 7.2. $R_0$ was positively correlated with ascertainment rates. Sensitivity analysis indicated that $R_0$ would become smaller with greater variance of the generation time.

**Discussion:** Estimated $R_0$ was broadly consistent with published estimate from endemic data, indicating that the vaccination coverage of 86% has to be satisfied to prevent the epidemic by means of mass vaccination. LHS was particularly useful in the setting of a refugee camp in which the background health status is poorly quantified.

Corresponding author
Hiroshi Nishiura,
nishiurah@gmail.com

## INTRODUCTION

Diphtheria, a bacterial disease caused by *Corynebacterium diphtheriae*, is a vaccine-preventable disease. Symptomatic patients initially complain sore throat and fever. Additionally, a gray or white patch causes the "croup," blocking the airway and causing a barking cough. Due to widespread use of diphtheria–tetanus–pertussis (DTP) vaccine

globally, the incidence has steadily declined over time, and thus, diphtheria is commonly perceived as a disease of pre-vaccination era. Nevertheless, sporadic cases and even epidemics of the disease have been yet reported especially in politically unstable areas, and many cases have been considered as arising from susceptible pockets of the vulnerable population (*Rusmil et al., 2015*; *Hosseinpoor et al., 2016*; *Sangal et al., 2017*).

In 2017, multiple diphtheria outbreaks were reported in refugee camps, including those in Yemen and Bangladesh (*World Health Organization (WHO), 2017a*). Of these, a Rohingya refugee camp in Bangladesh, which is temporarily located in Cox's Bazar, experienced a large-scale diphtheria epidemic. As of December 26, 2017, the cumulative number of 2,526 cases and 27 deaths were reported (*World Health Organization (WHO), 2017a*). To interrupt chains of transmission, emergency vaccination has been conducted among children since December 12, 2017, achieving the overall coverage greater than 90% by the end of 2017 (*World Health Organization (WHO), 2018*). Due to vaccination effort and other countermeasures, including contact tracing and hospital admission of cases, the epidemic has been brought under control, with incidence beginning to decline by the end of December 2017 (*World Health Organization (WHO), 2017a*).

Considering that diphtheria has become a rare disease in industrialized countries, epidemiological information on model parameters that govern the transmission dynamics has become very limited, and thus, it is valuable to assess how transmissible diphtheria would be through the analysis of the recent outbreak data. The basic reproduction number, $R_0$, is interpreted as the average number of secondary cases that are produced by a single primary case in a fully susceptible population, acting as the critical measure of the transmissibility. To date, an explicit epidemiological estimate of $R_0$ for diphtheria has been reported only by *Anderson & May (1982)*: using a static modeling approach to age-dependent incidence data with an assumption of the endemic equilibrium, $R_0$ was estimated as 6.6 in Pennsylvania, 1910s and 6.4 in Virginia and New York from 1934 to 1947. Subsequently, a few additional modeling studies of diphtheria took place (*Kolibo & Romaniuk, 2001*; *Sornbundit, Triampo & Modchang, 2017*; *Torrea, Ortega & Torera, 2017*), but none of these offered an empirical estimate of $R_0$.

Here we analyze the epidemiological dataset of diphtheria in Rohingya refugee camp, 2017, aiming to estimate $R_0$ in this particular epidemic setting. Given that the epidemic occurred among refugees, we explicitly account for uncertainties associated with unknown background information, including the fraction of previously immune individuals and ascertainment rate of cases.

## MATERIALS AND METHODS

### Epidemiological data

The latest epidemic curve was extracted from the report of the World Health Organization (WHO) Regional Office for South East Asia (SEARO) (*World Health Organization (WHO), 2017a*). Figure 1 shows the latest available epidemic curve. As of December 26, 2017 (the latest date of observation), a total of 2,526 cases have been reported. Cases consist of (i) confirmed cases: cases reported as positive for *C. diphtheria*e by multiplex assay, (ii) probable cases: cases with upper respiratory tract illness with laryngitis or

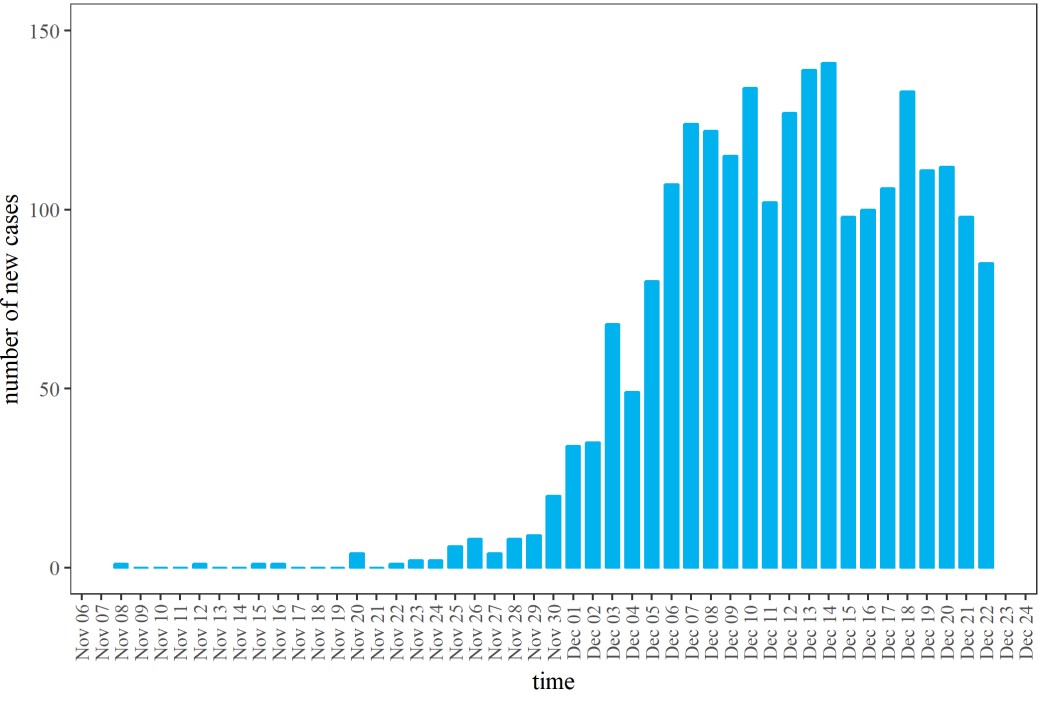

**Figure 1 Daily incidence of diphtheria cases in Rohingya refugee camp, 2017.** Daily number of new cases as extracted from the latest open data (*World Health Organization (WHO), 2017a*). The vertical axis represents the total of confirmed, probable and suspected cases. By December 11, 2017, the count represents suspected cases. On and after December 12, 2017, the case definition was improved, and probable cases replaced the majority.

nasopharyngitis or tonsillitis AND sore throat or difficulty swallowing and an adherent membrane (pseudomembrane) OR gross cervical lymphadenopathy, and (iii) suspected cases: any case with a clinical suspicion of diphtheria, including cases that are unclassified due to missing values (*World Health Organization (WHO), 2017b*). Prior to December 11, 2017, cases satisfied only the condition of suspected cases. The definition was improved on and after December 12 (*World Health Organization (WHO), 2017c*), enforcing to count mainly probable cases, while not restricting the reporting of suspected cases. For this reason, cases reported by December 11 are considered to have been likely over-ascertained compared with cases that were reported later under improved case definition. Mass child vaccination started on December 12, and according to the administrative coverage, greater than 90% vaccination coverage was achieved by December 30 (*World Health Organization (WHO), 2018*). Vaccine-induced immunity requires at least 7–14 days to become effective, and moreover, the reporting delay was assumed to be about four days (based on additional analyses; results not shown). For these reasons, the dataset from December 23–26 was discarded as the number of cases may be influenced by emergency vaccination and also biased by the reporting delay.

## Modeling methods

To formulate the epidemiological model, here we mathematically capture the epidemiological process of secondary transmission using $R_0$ and the serial interval,

i.e., the time from illness onset in the primary case to illness onset in the secondary case. We assume that secondary transmission does not take place before illness onset. According to a classical study by *Stocks (1930)* in the United Kingdom (UK), the time interval from first to second diphtheria cases in the household revealed a bimodal shape. Following *Klinkenberg & Nishiura (2011)*, the first peak corresponds to an independent infection in the community and the second peak reflects within-household transmission. As one of peak time-lags in the observed time interval was observed on day 8, we assumed that the mean serial interval was eight days, and we imposed an assumption that the coefficient of variation (CV) of the serial interval distribution was 50%, and later varied it from 25% to 75% as part of the sensitivity analysis.

To capture the epidemiological phenomena of reproduction, it has been shown that the renewal process can capture the serial stochastic dependence structure (*Nishiura, 2010*). Let $i_t$ be the number of new cases on day $t$. $g_\tau$ represents the distribution of the serial interval. To describe the time-dependent incidence $i_t$ on day $t$, we have

$$i_t = R_0 s_t \sum_{\tau=1}^{t-1} i_{t-\tau} g_\tau, \tag{1}$$

where $s_t$ represents the fraction of susceptible individuals on day $t$. The renewal process of this type is not original to the present study and has been applied to other settings including real-time epidemic modeling studies (*Asai & Nishiura, 2017*; *Dinh et al., 2016*; *Ejima & Nishiura, 2018*; *Endo & Nishiura, 2015*; *Nishiura et al., 2010, 2016*; *Tsuzuki et al., 2017*). It should be noted that the incidence $i_t$ includes both symptomatic and asymptomatic cases. Let $c_t$ be the reported number of cases on calendar day $t$. Supposing that only the fraction $\alpha_t$ among the total number of infections are diagnosed and reported, $c_t$ satisfies

$$i_t = \frac{c_t}{\alpha_t}, \tag{2}$$

where $\alpha_t$ is modeled as a function of $t$. Because the case definition was improved from December 12, 2017 onward, the ascertainment rate likely varied around that time. Namely, we set $\alpha_t = a_1$ for time by December 11 and $a_2$ on December 12 and later. We assumed that only the ascertainment rate changed as a function of time, and also that $R_0$ and depletion of susceptible individuals were unaffected by time.

We model the fraction susceptible $s_t$ on day $t$ in the following way. Let $\nu$ represent the previously immunized fraction so that only fraction $(1 - \nu)$ of the population is susceptible at the beginning of the epidemic. In addition to the previously immune fraction, $s_t$ decreases when natural infection takes place. Suppose that the total population size was $N$, $s_t$ is written as

$$s_t = 1 - \nu - \frac{\sum_{y=1}^{t-1} \frac{c_y}{\alpha_y}}{N}. \tag{3}$$

We assume that $N$ is equal to the population size of epidemic area within Rohingya refugee camp as 579,384 persons (*Banerji & Ahmed, 2017*). Accordingly, the renewal equation is written as

$$E(c_t; R_0, \nu, a_1, a_2) = \alpha_t R_0 \left( 1 - \nu - \frac{\sum\limits_{y=1}^{t-1} \frac{c_y}{\alpha_y}}{N} \right) \sum_{\tau=1}^{t-1} \frac{c_{t-\tau}}{\alpha_{t-\tau}} g_\tau \qquad (4)$$

where $\tau$ in the right-hand side indicates the time since infection (or the so-called "infection-age"). We assume that $c_t$ follows a Poisson distribution. The likelihood to estimate $\theta$ consisting of the parameters $R_0$, $\nu$, and $\alpha_t$ is derived as

$$L(\theta; \mathbf{c_T}) = \prod_{t=1}^{T} \left( \frac{E(c_t)^{c_t} \exp(-E(c_t))}{c_t!} \right), \qquad (5)$$

where $T$ is the latest time of observation (i.e., December 22 in our case study) and $\mathbf{c_T} = (c_1, c_2, \ldots, c_T)$.

## Uncertainty and sensitivity analyses

While we specified unknown parameters as $R_0$, $\nu$, and $\alpha_t$, it is expected that $R_0$ is correlated with initially immune fraction $\nu$ and also $\alpha_t$. Thus, it is vital to quantify $R_0$ while accounting for the uncertainty of other model parameters. In the present study, $\nu$ was estimated through uncertainty analysis, while $\alpha_t$ was estimated as a step function governed by two parameters, $a_1$ and $a_2$. Uncertainty in parameter values can be addressed by randomly sampling the uncertain parameter value from probability distributions (*Gilbert et al., 2014*). Here we use the Latin Hypercube sampling (LHS) method (*Sanchez & Blower, 1997*) in which a symmetric triangular distribution of $\nu$ was assumed to be in the range from 0.0 to 0.7; the health survey of Rohingya population indicated that overall 30.8% of children had received no vaccinations (*Guzek et al., 2017*), and the remaining 69.2% receives at least a single immunization, which is not necessarily against diphtheria, thus, vaccination coverage for diphtheria should be less than 69.2%. Thus, we expect that the actual coverage is nearby the mid-point of the range from 0.0 to 0.7. In addition, we examined the sensitivity of $R_0$ to variations in the length of the serial interval.

## Ethical considerations

The present study analyzed data that is publicly available. As such, the datasets used in our study were de-identified and fully anonymized in advance, and the analysis of publicly available data without identity information does not require ethical approval.

## RESULTS

Figure 2 shows univariate distributions of estimated parameters $R_0$ and $\alpha_t$ based on LHS ($n = 1,000$) of parameter $\nu$ from 0 to 0.70 with the peak at 0.35. $a_1$ Reflects ascertainment by December 11, 2017, while $a_2$ shows the same on and after December 12. $R_0$ took the minimum and maximum estimates at 4.7 and 14.8, respectively, with the median estimate

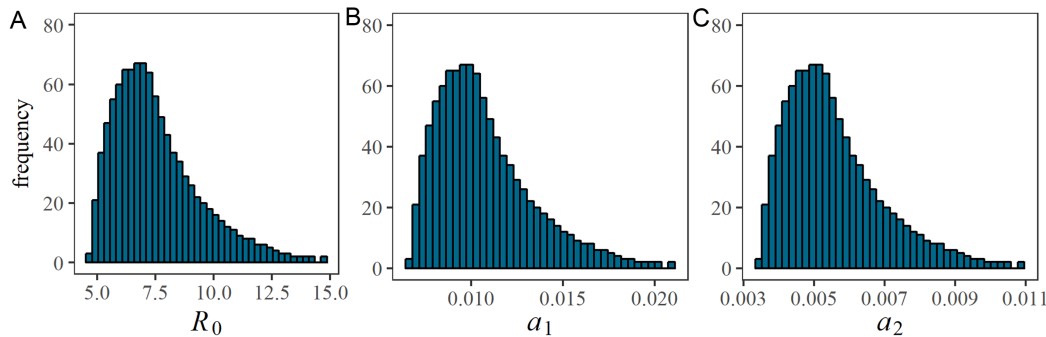

**Figure 2 Estimated values of the basic reproduction number and case ascertainment rate.** Univariate probability distribution of (A) the basic reproduction number, (B) $a_1$ by December 11 and (C) $a_2$ by December 12 from Latin Hypercube sampling ($n = 1,000$). During the Latin Hypercube sampling, the vaccination coverage, $\nu$, has a symmetric triangular distribution ranging from 0.0 to 0.7.

at 7.2. The distribution was skewed to the right with the mode at 6.7. Excluding lower and upper tails, 950 samples (95%) of $R_0$ (or what it can be assumed as the 95% tolerance intervals) were in the range of 5.0–12.3. Distributions of $a_1$ and $a_2$ were also right skewed. $a_1$ Ranged from 0.007 to 0.021 with the median 0.010, while $a_2$ ranged from 0.003 to 0.011 with the median 0.005. Lower and upper 95% tolerance intervals of $a_1$ and $a_2$ were (0.007, 0.017) and (0.004, 0.009), respectively. A decline of $\alpha_t$ was observed, because the incidence was supposed to have grown more during the early phase (i.e. by December 11) given the estimated epidemiological parameters. In practical sense, the timing coincided with the involvement of contact tracing practice enforced by the Médecins Sans Frontières.

Figure 3 shows the distributions of two estimated parameters in two-dimensional spaces and also the comparison between observed and predicted epidemic curve. As can be expected from Eq. (4), $R_0$ and $\nu$ were positively correlated given an epidemic curve. Specifically, as $\nu$ increases $R_0$ must also increase to achieve an identical epidemic curve. Similarly, $R_0$ was positively correlated with ascertainment rates, $a_1$ and $a_2$. While the ascertainment might have been lowered due to the stricter case definition, the small estimate of $a_2$ can also indicate that a substantial fraction of undiagnosed individuals existed and the susceptible fraction was then gradually depleted in the population. The observed and predicted epidemic curves are compared in the right lower panel of Fig. 3. While the model is kept simple with four unknown parameters, the predicted epidemic curve overall captured the observed pattern.

Sensitivity of $R_0$ to the variation in the serial interval is shown in Fig. 4. We varied the CV of the serial interval distribution from 25% to 75%. When the CV was 25%, the median and mode of $R_0$ from LHS were 9.4 and 8.2, respectively. When the CV was 75%, the median and mode of $R_0$ were estimated to be 5.7 and 5.2, respectively. That is, as the variance increases, the estimate of $R_0$ decreases. However, the variation in $R_0$ induced by the CV was smaller than the uncertainty associated with the initially immune fraction.

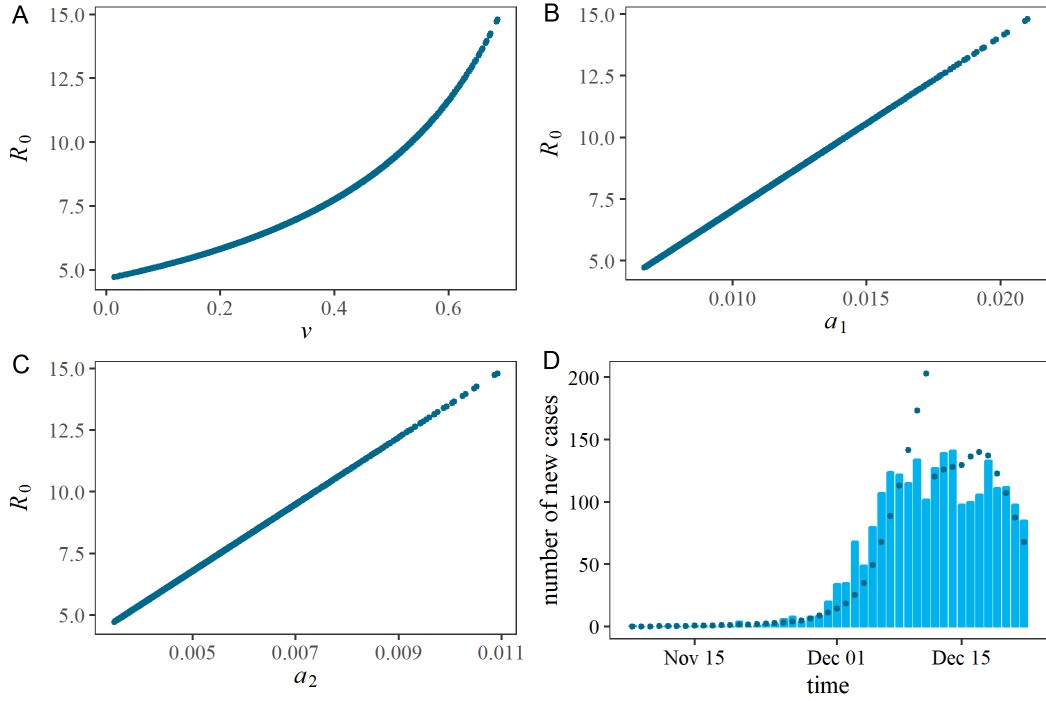

**Figure 3 Estimated correlations in each pair of estimated parameters, and comparison between observed and predicted epidemic curves.** (A), (B) and (C) represent a two-dimensional plot of estimated parameters. During the Latin Hypercube sampling ($n = 1,000$), the vaccination coverage, $v$, has a symmetric triangular distribution ranging from 0.0 to 0.7. (D) shows the comparison between observed and predicted epidemic curves. Bars constituting the epidemic curve show the observed data, while dots indicate predicted epidemic curve from Latin Hypercube sampling ($n = 1,000$).

## DISCUSSION

The present study estimated $R_0$ of diphtheria in the Rohingya refugee camp, explicitly accounting for case ascertainment and previously immune fraction. Since previously immune fraction $v$ of the refugee population was not precisely known, uncertainty analysis of $R_0$ was conducted with an input parameter assumption for $v$ employing the LHS method. $R_0$ ranged from 4.7 to 14.8 with the median estimate at 7.2. To our knowledge, the present study is the first to statistically estimate $R_0$ of diphtheria from epidemic data. For the statistical estimation of $R_0$, the renewal process model was employed, which has an advantage to handle the right-censored data during the course of an epidemic, compared with other available methods for completely observed data, e.g., *Nishiura (2010)*.

Estimated median $R_0$ was broadly consistent with the value ranging from 6 to 7 as indicated by *Anderson & May (1982)* based on a static model for endemic data that uses the age-dependent incidence in the UK. We have shown that the frequently quoted estimate agrees well with dynamically estimated $R_0$ from the refugee camp in the present day. Assuming $R_0 = 7$, to control diphtheria by means of mass vaccination, the coverage greater than 86% must be satisfied. Since our study focused on uncertainty and sensitivity analyses, the exact estimate of $R_0$ cannot be pointed out. However, despite the uncertainty

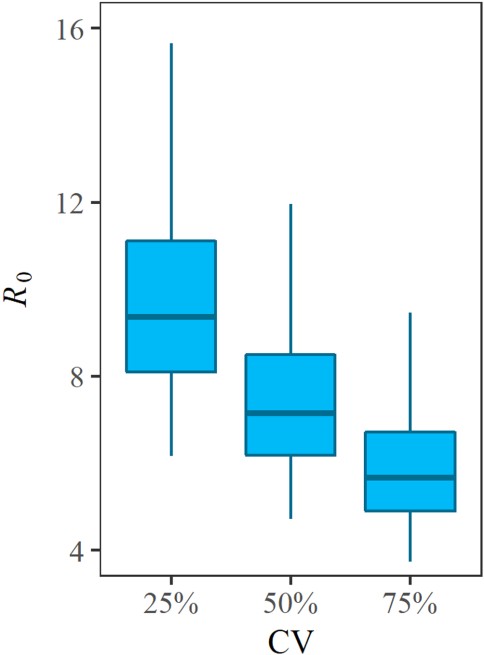

**Figure 4 Sensitivity of $R_0$ with respect to the serial interval.** $R_0$ was estimated with variable values of the coefficient of variations (CV) of the serial interval. Mean serial interval was fixed at eight days. Variations of $R_0$ along the vertical axis reflects the uncertainty associated with the initially immune fraction $v$ of the Rohingya refugee population.

regarding ν in this population, we estimated a distribution of $R_0$ consistent with previous estimates. While the mode of distribution for $R_0$ was 6.7, the validity of this value depends on the validity of our prior distribution of ν, which was not supported by any published evidence of this refugee population. Nevertheless, a Demographic and Health Survey data of the Rohingya population in Myanmar indicated a close value from 40% to 50% as the coverage of DTP (*Ministry of Health and Sports, 2017*). Ascertainment rates were jointly estimated only by using the epidemiological case data and the population size.

What we have shown in the present study is that when we have an access to not only the initial growth rate of the epidemic but also the incidence data around the time at which peak incidence is observed, $R_0$ and susceptible fraction can potentially be jointly quantified. Even without explicit estimate of the initially immune fraction, we have shown that an indication of the possible value of $R_0$ can be obtained through uncertainty analysis. LHS appeared to be particularly useful in the setting of refugee camp in which the background health status is not well quantified (*Helton & Davis, 2002*; *Nishiura et al., 2017*). LHS can offer probabilistic distribution of the outcome measure, $R_0$ in our case, and this method appeared to be particularly useful when one or more uncertain input information exist (*Elderd, Dukic & Dwyer, 2006*; *Coelho, Codeço & Struchiner, 2008*; *Samsuzzoha, Singh & Lucy, 2013*; *Gilbert et al., 2014*). While Bayesian modeling has replaced LHS to some extent of uncertainty analysis as it can also offer posterior distributions of even uncertain parameters (*Elderd, Dukic & Dwyer, 2006*;

*Coelho, Codeço & Struchiner, 2008*), there could be an issue of identifiability when two or more parameters are evidently correlated, e.g., as anticipated between $R_0$ and ν in our model (4). In such an instance, we cannot be sure if the limited epidemic data with the Bayesian estimation method can offer identifiable distributions for all parameters, and then LHS can remain to act as a useful tool for uncertainty analysis.

The estimated small ascertainment rate during the epidemic is considered as reflecting the time-dependent diagnostic practice at a local level. In fact, the small ascertainment rate can lead us to observing the low case fatality risk of diphtheria in the Cox's Bazar that has been estimated to be small (27/2,526 = 1%), although the right censoring would of course matter for the real-time interpretation (*Nishiura et al., 2009*; *Mizumoto et al., 2015*). Nevertheless, cases could have died in the community unnoticed with low specificity of the case definition, and this is endorsed as a possible reason for observing the small number of deaths by the *World Health Organization (WHO) (2017d)*. A follow-up study in this regard should be conducted in the future.

Several limitations must be noted. First, our model rested on a homogeneous mixing assumption. No heterogeneous patterns of transmission, including contact patterns and age-dependency were taken into account due to shortage of information. In addition, the time-dependent heterogeneity, including the reduced transmission potential due to contact tracing and rapid hospitalization, was not explicitly taken into account due to insufficiency of the data. If there were any additional indications or datasets that would allow explicit quantification of the effective reproduction number from December 12, that could give additional insights into the success of control measures. Second, for similar reasons, no spatial information was explicitly incorporated into the model. Third, a little more realistic features of refugee population, such as the impact of migration on the epidemic were unfortunately discarded in the present study. Similarly, one could investigate how overcrowding and malnutrition in the deprived population would help enhance the spread of diphtheria, given sufficient data backup from epidemiological investigations.

While these features need to be explicitly quantified in the future, we believe that our study adds an important piece of evidence to the literature on diphtheria. The transmissibility of diphtheria in the refugee population was estimated to be consistent with that in an endemic setting and mass vaccination must satisfy at least the coverage of 86% to halt the major epidemic of diphtheria.

## CONCLUSION

The present study estimated $R_0$ of diphtheria in the Rohingya refugee camp, explicitly accounting for case ascertainment and previously immune fraction. Since previously immune fraction ν of the refugee population was not precisely known, uncertainty analysis of $R_0$ was conducted with an input parameter assumption for ν employing the LHS. $R_0$ ranged from 4.7 to 14.8 with the median estimate at 7.2. LHS can offer probabilistic distribution of the outcome measure, and this method appeared to be particularly useful in the setting of refugee camp in which the background health status is poorly quantified.

### Funding

The present study was financially supported by the Japanese Society for the Promotion of Science (JSPS) Grant-in-Aid for Scientific Research (KAKENHI) numbers for Hiroshi Nishiura: 16KT0130, 16K15356, and 17H04701, Ryota Matsuyama with 16H06581, and Shinya Tsuzuki with 17H06487. Hiroshi Nishiura received funding from the Japan Agency for Medical Research and Development, JSPS Program for Advancing Strategic International Networks to Accelerate the Circulation of Talented Researchers, The Telecommunications Advancement Foundation, Inamori Foundation, and the Japan Science and Technology Agency (JST) Core Research for Evolutional Science and Technology (CREST) program (JPMJCR1413). Akira Endo was supported by The Nakajima Foundation. The funders had no role in study design, data collection and analysis, decision to publish, or preparation of the manuscript.

### Grant Disclosures

The following grant information was disclosed by the authors:
Japanese Society for the Promotion of Science (JSPS) Grant-in-Aid for Scientific Research (KAKENHI): 16KT0130, 16K15356, 17H04701, 16H06581, 17H06487.
Japan Agency for Medical Research and Development.
JSPS Program for Advancing Strategic International Networks to Accelerate the Circulation of Talented Researchers.
The Telecommunications Advancement Foundation.
Inamori Foundation.
Japan Science and Technology Agency (JST) Core Research for Evolutional Science and Technology (CREST) program: JPMJCR1413.
The Nakajima Foundation.

### Competing Interests

Hiroshi Nishiura is an Academic Editor for PeerJ.

### Author Contributions

- Ryota Matsuyama conceived and designed the experiments, performed the experiments, analyzed the data, contributed reagents/materials/analysis tools, prepared figures and/or tables, authored or reviewed drafts of the paper, approved the final draft.
- Andrei R. Akhmetzhanov performed the experiments, analyzed the data, contributed reagents/materials/analysis tools, authored or reviewed drafts of the paper, approved the final draft.
- Akira Endo performed the experiments, analyzed the data, authored or reviewed drafts of the paper, approved the final draft.
- Hyojung Lee performed the experiments, analyzed the data, contributed reagents/materials/analysis tools, prepared figures and/or tables, authored or reviewed drafts of the paper, approved the final draft.

- Takayuki Yamaguchi performed the experiments, authored or reviewed drafts of the paper, approved the final draft.
- Shinya Tsuzuki performed the experiments, authored or reviewed drafts of the paper, approved the final draft.
- Hiroshi Nishiura conceived and designed the experiments, performed the experiments, analyzed the data, contributed reagents/materials/analysis tools, prepared figures and/or tables, authored or reviewed drafts of the paper, approved the final draft.

## Supplemental Information

Supplemental information for this article can be found online at http://dx.doi.org/10.7717/peerj.4583#supplemental-information.

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
