# Peer review of "Uncertainty and sensitivity analysis of the basic reproduction number of diphtheria: a case study of a Rohingya refugee camp in Bangladesh, November–December 2017"

_PeerJ, doi:10.7717/peerj.4583_

## Round 0.1 · original submission · Major Revisions

You have comments from two peer reviewers, of which one is public (i.e., not anonymous).

A few comments:

- Although the decision is "major revisions", overall the comments are not especially severe, and (without making any commitments) I expect that you can satisfy the referees.

- I expect a response document that addresses each and every point raised by the referees. However, my standard for "addressing" does not mean that you must slavishly adhere to every point.

In some cases, addressing a point may mean explaining why you chose not to implement a particular suggestion. Needless to add, I don't recommend this for all the suggestions (I will rely on the input of the same referees for the revised version).

Reviewer 1 ·

Basic reporting

The authors present an interesting and well-structured paper, yet there are several areas where additional refinement is required.

Throughout the paper there is writing that can be made more concise and clear. At points, the writing needs substantial clarification. For example, the description of the case definitions and their changes was not clear –the case definition after 11 Dec was detailed, but it is not stated what the definition was prior. Additionally, correct punctuation is missing throughout. For example, a comma [“,”] should often be used before the word “including” when you are listing examples, etc. (see lines 62, 82, 246

Literature citations are generally sufficient throughout, with the exception of one key citation. The authors present their model, yet provide no background on its previous use, reason for use, etc. They also do not explicitly state that they developed this model, nor do they provide any defined validation of this model. Upon searching, I found a previous paper proposing this model, by the senior author (Nishiura Int. J. Environ. Res. Public Health 2010). The author should revise the methods to reference this work, and explain why this model is being used in this context instead of other options, such as the epidemic growth model.

Experimental design

This article presents original primary research that is within the Aims and Scope of PeerJ. The research question, that of estimating the R0 for diphtheria in a modern context, is relevant and meaningful, and fills an important gap in knowledge.
The analysis performed is of high technical and ethical standards, however there are some major flaws in the design and/or description and purpose of the authors’ methodological choices.

The first problem is the general description of the methods and layout of the methods section. As presented, they are not well structured and lead to substantial confusion and questions. The methods start off with a definition of 2 parameters, i and g, then lead into a lengthy description of g. This is before we are presented with any model or basic framework. This type of information should go after the model is presented.
Next, the readers are given the model, but without any explanation of the choice of the model and derivation of it. As stated above, this renewal process model is defined in the methods, though there is no background provided on why this model is being used, how it was developed, how it is valid, etc. As is reads, it is unknown whether the authors developed this model for this work or they believe that this is a well-known model that they are just applying. In either case, this should be stated. From what I found in the literature, they did not develop this model for this work, as it was previously presented by Nishiura in 2010.

Another major problem is their choice of prior distributions for unknown parameters, and their descriptions of these. As they state, they defined the sampling distribution for v, the vaccination coverage, with a triangular distribution from 0-0.7 (line 149). However, they also state in the same sentence that a previous survey found that 30.8% of children had received no vaccinations (line 150), thus according to that survey, 69.2% had received vaccination. Given that it is well documented that diphtheria has among the highest coverage rates of vaccines, it can be presumed that if this survey is correct, I would have expected the authors to set a prior for v such that the mean prior distribution is at ~70%. Instead, their prior distribution has a mean of 35%, and does not allow v to go above 70%, which is possible, given vaccination coverage among adults may be substantially higher than that among the surveyed refugee children.

Additional to the problem with the distribution allowed for v, there is no mention of what distributions were allowed for alpha1 and alpha2. It can be assumed they used LHS to determine the values for these parameters along with v, but there is no mention of what values or distributions were used. In general, the authors need to provide a clear and concise description of the input variables and parameter values/distributions inputted into the model. This should be noted in the methods, and it would be useful to have a table of this information.

Validity of the findings

There is no mention in the Results section of the estimation results for v. This is actually presented in the discussion, though it is not detailed (i.e., no range, confidence intervals, etc.; line 221). At the very least, this should be moved to the results and additional information should be added.

I also do not understand how estimates of alpha1 and alpha2 were so different, particularly given there was no dramatic change in the number of reported diphtheria cases around December 11. If these parameters were as different as these estimated values say, there should have been a notable change in the epidemic curve, demonstrating a district change in case ascertainment. Furthermore, the hyperbolic relationship between alpha1 and R0 does not make sense, and the authors’ reasoning that it occurs because alpha1 crosses 1 does not justify this relationship (line 177). Why would alpha1 go from 0.02 (i.e., ascertainment of 1/50 cases) at R0=8 to >1 at R0>8.5? It seems that there are some misspecification sampling issues occurring.

I have some concerns that the problems with the parameter distributions state in the “Experimental Design” section of this review may be influencing the results of this study. Specifically, the finding that the median R0 for this outbreak was 5.8, which correlates directly to the mean/median/mode of the v input distribution of 35%, given my above concerns that this is a misspecified prior (given prior knowledge from both the survey and DHS). This may very well be a correct estimate, but I would like to see this analysis redone with a more appropriate set of parameter priors.

Additionally, I would have liked to see a brief mention or discussion of the possibility that case ascertainment of the last several days/weeks of the case data may be inaccurate due to reporting delays. It is highly likely that in a setting such as a refugee camp, and especially in an outbreak as such, that reporting of cases would be delayed substantially.

Additional comments

This is an interesting and timely analysis. I believe if these problems detailed here can be addressed, it will provide useful additional information on diphtheria in the literature. I have provided some additional grammatical edits and minor comments in the PDF of the manuscript, as well as these listed below. I would recommend spending some time to clarify some confusing sentences, make the writing more concise, and recheck for grammatical errors.

Specific comments:
Line 45. Please revise this to be more concise and clear. Additionally, the grey or white patch causes the "croup".

Line 102. It needs to be made clearer what data times and number of cases were included in this analysis. Please state this all together.

Line 114. As Nishiura is an author, these seems like an invalid reference. Please reference previous use of this, or use a brief explanation of it.

Line 132. This is confusing. Is this N=579,384 just the epidemic area within the refugee camp, or is this the population of the entire refugee camp. Also, this can be rewritten more concisely.

Line 135. What does tau represent in equation 4? I found it in Nishiura 2010 representing time since infection. Is this correct here too? This needs to be described.

Line 194. The authors state that: “This dependency [R0 and alpha2] is anticipated, because R0 would influence how many susceptibles to be depleted to curb the epidemic curve and that is regulated by the value of a2.” Why is this regulated by alpha2? Alpha is a parameter of case ascertainment and does not have any bearing on the disease process. Please clarify this.
Line 225. The authors state that “It is remarkable that ascertainment rates were jointly estimated only by using the epidemiological case data and the population size.” I would argue that this is not necessarily remarkable given that ascertainment rates were not actually estimated with any accuracy. Then presented estimates of alpha1 and alpha2 are qualitatively different, despite lack of evidence for this in the epidemic curve or other knowledge. Additionally, the confidence ranges for these estimates are enormous.

Line 229. I would be very hesitant to state that this is the peak given these data are so recently reported. In outbreak and refugee settings, it is very common that large reporting delays occur, so it is possible that the leveling-off of cases is merely a factor or reporting delay.

Figure 2. Because alpha1 and alpha2 are reporting proportions that can be greater than or less than 1, these axes should be reported in log-scale.

Annotated reviews are not available for download in order to protect the identity of reviewers who chose to remain anonymous.

·

Basic reporting

The basic reporting is well done. You may add some references or sources of information on the vaccination campaign, for instance how do we know that the coverage reached was 90% (administrative coverage or survey?).

Experimental design

Well done

Validity of the findings

Constant Ro :
You state in the introduction (line 62) that “Due to vaccination effort and other countermeasures including contact tracing and hospital admission of cases, the epidemic has been considered to be gradually brought under control ….”. In the methods (Line 101), you justify why you have not include the effect of the vaccination campaign in your model. But the effects of the other countermeasures are not discussed. While vaccination does affect only the effective reproductive ratio, contact tracing, rapid hospitalisation and prophylaxis given to the contacts may have changed the basic reproductive ratio. The assumption of a constant Ro overtime needs to be discussed. Furthermore it would be interesting to explore a varying Ro, as you have included 2 ascertainment rates, would it be possible to include 2 Ro. Knowing that the Ro was constant or changed overtime has some implication in your estimate, and it would inform the public health actors in the best strategy to control outbreaks (vaccination, prophylaxis, treatment and isolation).

Low CFR and partial immunization :
Compared to other epidemics (Recent Outbreaks of Diphtheria in Dibrugarh District, Assam, India; Diphtheria outbreak in Lao People's Democratic Republic, 2012-2013; Diphtheria outbreak with high mortality in northeastern Nigeria) the outbreak in Cox bazar had a much lower Case Fatality Ratio (27/2526 =1%). A number of reasons could explain this: a good case management (this is unlikely in this setting at this stage of the emergency), deaths were not recorded either because they died after discharge (they kept them hospitalized 48 hours only, which is short for diphtheria) or they died in the community unnoticed, low specificity of the case definition and low fatality of people with partial immunization (not having received the complete course of vaccinations, see “Epidemic Diphtheria in the Republic of Georgia, 1993–1996: Risk Factors for Fatal Outcome among Hospitalized Patients”). The latter reason is likely in this setting, I wonder if this would affect the model; It might deserve to be discussed.

Additional comments

It is an interesting article, that would be useful for health actors.

---

## Round 0.2 · accepted · Accept

As your revisions have satisfied both of the reviewers and me, I am happy to accept your revised manuscript. One of the reviewers has supplied an annotated PDF with a number of very small suggested changes.

While your manuscript is accepted without conditions, I do urge you to implement these suggested changes, which will improve the readability of your paper.

# Reviewer 1 ·

Basic reporting

The author has sufficiently improved the manuscript from the the previous version. It is currently up to the standards of PeerJ for publication.

Experimental design

No comment

Validity of the findings

After the revisions and redo of the analysis, the findings appear to be valid.

Additional comments

Te edits have made this paper much easier to read and fixed validity issues. Very nice work. I have attached an annotated PDF with a couple additional minor edits.

Annotated reviews are not available for download in order to protect the identity of reviewers who chose to remain anonymous.

·

Basic reporting

No comments

Experimental design

No comments

Validity of the findings

No comments

Additional comments

The authors well responded to the comments of the first revision.